# Nicotinic Receptors in Human Chromaffin Cells: Characterization, Functional and Physical Interactions between Subtypes and Regulation

**DOI:** 10.3390/ijms25042304

**Published:** 2024-02-15

**Authors:** Amanda Jiménez-Pompa, Almudena Albillos

**Affiliations:** Departamento de Farmacología y Terapéutica, Universidad Autónoma de Madrid, 4 Arzobispo Morcillo Str., 28029 Madrid, Spain; a.jimenezpompa@gmail.com

**Keywords:** nicotinic receptor, α7 subtype, α3β4 subtype, patch-clamp, fluorescence, FRET, acetylcholine, choline, Ca^2+^

## Abstract

This review summarizes our research on nicotinic acetylcholine receptors in human chromaffin cells. Limited research has been conducted in this field on human tissue, primarily due to the difficulties associated with obtaining human cells. Receptor subtypes were characterized here using molecular biology and electrophysiological patch-clamp techniques. However, the most significant aspect of this study refers to the cross-talk between the two main subtypes identified in these cells, the α7- and α3β4* subtypes, aiming to avoid their desensitization. The article also reviews other aspects, including the regulation of their expression, function or physical interaction by choline, Ca^2+^, and tyrosine and serine/threonine phosphatases. Additionally, the influence of sex on their expression is also discussed.

## 1. Introduction

Nicotinic acetylcholine receptors (nAChRs) are ligand-gated cationic channels formed by different nAChR subunits, α1–α7, α9, α10, β1–αβ4, γ, δ, and ε in mammals, that assemble into pentamers to constitute a variety of nAChR subtypes. The different combination of subunits confers specific pharmacological and kinetic properties to each receptor subtype [1], which are generally divided into two large groups: homomeric (homopentamers formed by subunits α equal to each other: α7–α9) and heteromeric (formed by a combination of α and β subunits: α2–α6 and β2–β4) [2].

The binding of a neurotransmitter to one or more orthosteric sites induces a conformational change in the ligand-gated ion channel, allowing the flux of ions for which they are selective (Cl^−^, Na^+^ and/or Ca^2+^). This activation generates very fast synaptic transmissions (on the order of milliseconds) in smooth and skeletal muscle, in the central nervous system and peripheral nervous system, and in the endocrine system. It can also trigger slower processes involving Ca^2+^ and other second messengers in non-excitable cells, such as T lymphocytes, glia, or endothelial cells [2].

nAChRs are ubiquitously and heterogeneously expressed in different regions of the central and peripheral nervous systems. Furthermore, some homomeric subtypes such as the α7 are also expressed in non-neuronal tissues and cells such as microglia, astrocytes and oligodendrocytes, keratinocytes, platelets, cells of the immune system, tumor cells, and epithelia (intestinal, pulmonary, and oral) [3].

Receptors containing α4, β2, and α7 subunits are highly expressed in the brain, although their distribution in different regions varies greatly depending on the species [1,2,4]. In the α4β2 subtype, the stoichiometry between α and β subunits determines the sensitivity to the agonist and the permeability to Ca^2+^. It is directly involved in addiction and dependence phenomena due to its abundant expression in the dopaminergic reward circuit.

The α7 nAChR subtype (-nAChR) is highly expressed in regions involved in cognitive functions and related to memory and synaptic plasticity, such as the hippocampus, the cortex and subcortical areas of the limbic system [5]. Its ionotropic function in presynaptic locations is fundamental for Ca^2+^-dependent nervous transmission [6], as well as its metabotropic function through its binding to G proteins, which play fundamental signaling and neuroprotective roles in the brain [7]. Although most α7 subunits form homomeric receptors, heteromeric α7β2 receptors have recently been described in the human species in the cerebral cortex [8,9,10], as well as α7dupα7, an aberrant subunit of α7, which limits the function of the receptor [11,12].

The expression of α2 subunits is one of the most variable depending on the species. In humans, it is expressed together with β2 in area 21, which belongs to Wernicke’s area [1]. The genes that encode subunits α3, β4, and α5 form a cluster in human chromosome 15, which indicates the close relationship that exists between them. Receptors containing the α3 subunit are the most expressed in the peripheral nervous system, with α3β4 being the main receptor involved in ganglionic synaptic transmission. In the brain, its expression is restricted to certain neuronal areas of the hippocampus with specific functions [13,14], as well as in the cerebellum and higher nuclei. The inclusion of the α5 subunit increases its permeability to Ca^2+^, affinity for agonists, and desensitization of the receptor [15].

nAChRs with α6 subunits are expressed primarily in the ventral tegmental area and other structures of the basal ganglia [16] and are involved in cognitive processes. Given their role in regulating dopamine release, they have been proposed as a therapeutic target for the treatment of Parkinson’s disease [17]. They have also been found in structures related to vision, forming heteromeric receptors with different α and β subunits [1].

The α9 subunit, mostly expressed in the immune system, can form homomeric and heteromeric receptors together with α10. Its location in the nervous system is very restricted, and in recent years, it has been implicated in the perception of pain, being investigated as a potential therapeutic target to alleviate it [18]. In addition, in rats, nAChRs containing α9 subunits have been described in the adrenal medulla and have been shown to contribute to the adaptation of this tissue to stress situations [19].

## 2. nAChRs in Chromaffin Cells of the Adrenal Gland

The adrenal glands are endocrine organs that are located on the kidneys and are formed by two structurally and functionally well-differentiated areas: the cortex, of mesodermal origin, which mainly produces three types of hormones, mineralcorticoids, glucocorticoids, and androgens, and the medulla, derived from the neural crest, whose main secretion product is catecholamines (dopamine, adrenaline, and noradrenaline) [20]. As a whole, this organ is responsible for the systemic response to stress.

The adrenal medulla, where chromaffin cells reside, is considered a large specialized ganglion. It is innervated by sympathetic preganglionic fibers of the splanchnic nerve, whose somata are located in the mediolateral horn of the thoracic region of the spinal cord [21]. These neurons project their axons through the ganglia of the paravertebral sympathetic chain, leave the celiac ganglion without synapsing, and reach the adrenal capsule (Figure 1). The fibers penetrate the gland by crossing the cortex, and their axonal endings directly contact one or several chromaffin cells [22], which act as postsynaptic neurons that, instead of innervating a tissue and releasing neurotransmitters locally in it, secrete their stored products directly into the bloodstream.

The Ca^2+^-dependent synaptic connection between splanchnic nerve terminals and chromaffin cells is known as “excitation-secretion coupling”, a term adapted by Douglas and Rubin (1961) [23] from the one coined by Sandow in 1952 [24] after their observations in skeletal muscle. The acetylcholine (ACh) released by the splanchnic nerve terminals binds to nicotinic (and muscarinic) receptors on the chromaffin cell, activating them. The nAChRs change their conformation by opening the ionic pore, allowing mainly Na^+^ ions to pass through, but also Ca^2+^ in variable proportions depending on the receptor subtype. This entry of positive charges produces a depolarization in the cell membrane, which in turn leads to the activation of voltage-gated Na^+^ channels. The entry of Na^+^ through these channels depolarizes the plasma membrane, increasing the probability of voltage-gated Ca^2+^ channels opening. This entry of Ca^2+^, which also induces the release of Ca^2+^ itself stored in intracellular stores, is ultimately responsible for the fusion of mature secretory vesicles with the plasma membrane through a process of exocytosis, releasing their content of catecholamines into the bloodstream [25] (Figure 2).

Chromaffin cells share a large number of functional characteristics with sympathetic neurons because they have the same embryonic origin. Thus, they have synaptic or secretory vesicles and nicotinic [26,27,28,29] and muscarinic receptors [30] on which ACh acts, released by the splanchnic nerve. They fire action potentials [31,32,33]; have Ca^2+^, Na^+^, and K^+^ voltage-dependent channels [34,35]; present nitric oxide regulatory mechanisms [36]; and have neuronal growth factors such as NGF, TGF, and GDNF [37,38], in addition to having receptors for many other neurotransmitters such as GABA, ATP, opioids, and neuropeptides [39,40]. For all these reasons, in addition to the ease of obtaining and maintaining them in culture, chromaffin cells have frequently been used as a model for the study of basic mechanisms of neurophysiology, regulated secretion, and pharmacology [41,42].

### 2.1. Non-Human Species

Extensive research has been performed in chromaffin cells of non-human species in order to investigate their nAChR subunit composition. In bovine chromaffin cells, α3, α5, and β4 but not β2 subunits were detected by the reverse transcription-PCR analysis of mRNA from adrenal medulla [43]. Also, the α7 gene was successfully cloned [44], and cell surface expression of α7-nAChRs was determined using the α7-nAChR antagonist, the α-bungarotoxin (α-Bgtx), in binding experiments [45,46], together with antibody detection [47]. Moreover, functional studies using this toxin showed a function of the α7-nAChR in bovine chromaffin or PC12 cells [48,49,50], but in some other experiments, catecholamine secretion was not inhibited by the toxin [51,52,53,54].

In rat chromaffin cells, the most frequently encountered receptors comprised α3β4 and α3β2 with the addition of α5 subunits according to the study performed by Di Angelantonio and colleagues using RT-PCR and immunocytochemistry analysis [55]. Recently, Hone and colleagues [56] showed that these cells express two main heteromeric subtypes, namely α3β2β4- and α3β4-nAChRs, using the novel α-conopeptide PeIA-5469 that targets α3β2-nAChRs; the α-conotoxin (α-Ctx) TxID, a selective antagonist of α3β4-nAChRs [57,58]; and positive allosteric modulators for α4β2-, α4β4-, and α7-nAChRs. These subtypes are expressed by two populations of chromaffin cells: one population expresses α3β4 and the other one expresses both α3β2β4 and α3β4 subtypes. Also, the detection of α7-nAChRs was supported by α7 mRNA identification in rat chromaffin cells [19,55,59,60]. Inward currents induced by nicotine pulses were found to be unresponsive to α-Bgtx and low doses of methyllycaconitine (MLA) [55]. However, Hone and colleagues [56] recently showed that functional α7-nAChRs are expressed in rat adrenal chromaffin cells, determined by three α7-selective ligands: the agonist PNU282987, the positive allosteric modulator PNU120596, and the antagonist α-Ctx [V11L,V16D]ArIB.

### 2.2. Human Species

In human chromaffin cells (HCCs), Mousavi and colleagues detected in 2001 the presence of messenger RNAs (mRNA) of the seven subunits investigated (α3, α4, α5, α7, β2, β3, and β4) in the adrenal medulla of a 42-year-old donor [60]. Later on, we described the presence of functional α7-nAChRs using electrophysiological techniques and α7-nAChRs antagonists such as α-BgTx (1 μM) and MLA (10 μM) [61]. It is very relevant to mention that the expression of functional α7-nAChRs increases with repeated stimulation with ACh or another agonist that completely and simultaneously activates the α7- and α3β4*-nAChRs ([62], see below). In addition, an almost full block of the β4*-nAChRs currents and exocytosis (99.6% and 94%, respectively) was achieved in these cells using the α-Ctx BuIA (T5A,P60) (BuIA, selective for β4*-nAChRs; [63]) [64]. Subsequently, we confirmed the presence of α7, α3, and β4 subunit mRNAs both in the adrenal medulla and in isolated HCCs [65].

To further characterize the non-α7-nAChRs in HCCs, electrophysiological assays were performed, and specific α-Ctxs were employed. Besides the nicotinic current block by BuIA, the use of α-Ctx LvIA(N9R,V10A) (LvIA, selective for β2*-nAChRs; [65]) and α-Ctx TxID, confirmed that the main non-α7-nAChR in HCCs is the α3β4*-nAChR (block of 98% and 99% of the nAChR-elicited currents by BuIA and TxID, respectively), to which a minority population of β2*-nAChRs is added (block by LvIA of 7%). In contrast, α-Ctx PeIA(A7V,S90H,V10A,N11R,E14A) (selective of α6*-nAChRs, [65]) was only effective at very high concentrations (above 100 μM), suggesting the low presence of these subunits in HCCs [58,65]. All these studies lead to the conclusion that the main functional nAChRs expressed by HCCs are the α3β4*- and α7-nAChRs, accompanied by α5 and β2 regulatory subunits. 

Regarding the contribution to the exocytotic process, we found that α7-nAChRs currents alone did not trigger exocytosis, but the depolarization induced by these currents could elicit it [61]. In contrast, the current passing through the α3β4* nAChR ionophore was able to evoke exocytosis by itself or by triggering depolarization [64,66].

As we explained in the introduction, at least two different nAChRs subtypes are expressed in the different tissues. The possibility of functional and physical interaction between different subtypes has been unexplored until now. We found it to be of interest to investigate it in a human cell that express mainly two of these receptors, making it easier to find out the mechanism of interaction between them and its consequences. We designed the following experiments to investigate it [62].

First, to assess the functional interaction between α7- and non-α7-nAChRs subtypes, we stimulated HCCs using successive pulses of ACh (potentiation protocol) and recorded the elicited currents using the perforated-patch configuration of the “patch-clamp” technique in the voltage-clamp mode. This protocol resulted in the potentiation of the nicotinic current in both amplitude and charge, until the maximum effect was reached (Figure 3).

Another aspect observed with this protocol was the decrease in the activation time of the currents with each successive pulse, which might be due to the contribution of α7-nAChRs, characterized by their fast activation and inactivation kinetics and their crucial role in rapid synaptic transmission [67,68,69,70]. To evaluate this hypothesis, we applied, before and after the ACh potentiation protocol, pulses of choline (Chol), a full and partial agonist of α7- and α3β4-nAChRs, respectively, to record the activity of the α7-nAChR reflected in the fast peak current transient elicited by this agonist [61]. After successive pulses of Chol, the desensitization of the α7-nAChRs occurred, decreasing the evoked current. However, when the potentiation protocol was performed between two Chol pulses, the current of the second pulse of Chol was enhanced by almost 3 times compared to the initial current (Figure 4B,C “ACh+”). These results suggest that the full activation of the α3β4-nAChRs is essential to prevent the desensitization of the α7-nAChRs, something that physiologically occurs with ACh (but not with Chol, a partial α3β4-nAChRs agonist).

Taking into account the important effect that the co-activation of α3β4-nAChRs together with α7-nAChRs has on the activity of the latter subtype in HCCs, we wanted to check if the reverse could also be observed, that is, whether α7-nAChRs activity somehow modulated α3β4-nAChRs function. To carry this out, we design two approaches. (1) The first approach was the desensitization of α7-nAChRs through the application of successive Chol pulses. Once they were desensitized, we performed the usual potentiation protocol applying successive pulses of Ach. (2) The second approach was the block of α7-nAChRs via perfusion with α-Ctx ArIB, an α7-nAChRs selective blocker, before and throughout the potentiation protocol. In both cases, the peak current did not increase after the successive pulses but rather decreased.

The Ca^2+^ dependence of this functional interaction was investigated. It has been described that the activation of α7-nAChRs in the somatic spines of neurons induces rapid Ca^2+^-dependent trafficking of these same receptors through secretion mediated by SNARE proteins [71]. In our model, removing Ca^2+^ from the extracellular solution leads to the abolition of the ACh-elicited nicotinic current potentiation, which might suggest that this process and the overexpression of nAChRs in the membrane occur by exocytosis. However, even in the absence of Ca^2+^, the full activation of both receptor subtypes is capable of potentiating the α7-nAChRs current. Thus, the increase in activity and expression of the α7-nAChRs must be regulated, in part, by another mechanism independent of extracellular Ca^2+^.

One of the first questions that arises in light of these results is whether the increase in nicotinic current in HCCs is due to a greater expression of nAChRs in the plasma membrane [72] or to the fact that those receptors already available acquire a more easily activatable conformation [73]. The first process occurs on a scale of seconds, and therefore, it can be ruled out that the potentiation is given by an increase in the de novo synthesis of these proteins. Labeling experiments with subtype-selective α-Ctx linked to Alexa fluorophores show that after ACh potentiation, there is an increase in the expression of both α7- and α3β4-nAChRs. In the same way as in activity experiments, this increase does not occur if one of the two subtypes is desensitized or blocked, which reinforces the idea of the joint modulation of these two subtypes. The overexpression of nAChRs mediated by prolonged incubation of a nicotinic agonist such as nicotine itself is a process that had already been described previously [74], but the particularity of our finding is that the increase in expression occurs immediately after the application of a series of short pulses of ACh. Together, these results indicate that for potentiation and maintenance of a stable nicotinic current in HCCs, α3β4-nAChRs also require complete and functional activation of α7-nAChRs.

In order to check if there was physical interaction between α7- and α3β4*-nAChRs (proximity in the receptors between 10–100 Å), we performed the photobleaching technique of a small area of interest with the Cy3 marker (acceptor) to measure the efficiency of FRET. The results, even with those pharmacological maneuvers that decreased the fluorescence intensity in the expression of nAChRs, were positive in all cases, showing that (i) indeed, there is physical interaction between α7-nAChRs and α3β4*-nAChRs in HCCs and (ii) this interaction is independent of activity.

The question arises as to whether this interaction responds to the possible conformation of a mixed heteromeric receptor or to the interaction of independent receptors in a single functional unit. The α7-type subunits predominantly form homomeric receptors; however, it has been seen that they can also form functional heteromeric receptors by combining stably with β3, β2, and β4 subunits [75,76,77] in heterologous expression systems such as *Xenopus* oocytes and have even been able to be expressed stably and functionally in the primary culture of bovine chromaffin cells [77]. The combination with β2 has also been found and described in neurons of the cerebral cortex [8,9]. In bovine chromaffin cells, the possibility of a mixed α3α7β4 heteromeric receptor has already been previously discussed upon observing the notable kinetic differences of the nicotinic currents evoked in these cells with respect to those obtained in *Xenopus* oocytes that expressed α7- and α3β4-nAChRs independently [78,79].

The possible formation of a mixed α7 and non-α7 nAChR would be of great interest. At a physiological level, the agonist binding sites would be modified [80], which would confer different kinetic and functional properties to this new receptor. Given the low efficiency of the traffic to the plasma membrane of the α7 subtype, the possible interaction with other subunits could also facilitate this process, thus increasing its presence in the membrane [81,82,83]. The inclusion of an α5 subunit, as well as a β3, increases the sensitivity and desensitization of the heteromeric α3β4 and α3β2 receptors [15,84,85,86], as also occurs with the α4β2 subtype [87,88].

It cannot be ruled out, however, given the wide variety of tissues in which α7-nAChRs are found colocalizing with other nAChRs [1,89,90], that another possibility that would explain our results is reversible receptor–receptor binding. This type of junction has already been described in the peripheral nervous system [91,92], where, due to the interaction of α3 and β4 subunits with scaffolding proteins such as PSD95, functional groups or clusters of α3β4-nAChRs that enhance signal transmission are formed. In the hippocampus, α7-nAChRs form this type of cluster in GABAergic interneurons, which is positively regulated by neuroligins, neurotrophins, and increases in NMDA receptor activity [93,94,95] and negatively regulated by the scaffolding protein PICK1 [96]. In the ciliary ganglia, it would be the microdomains formed in the lipid rafts that would regulate this aggregation, while in muscle receptors, it would be regulated by phosphorylation/dephosphorylation processes dependent on Src family enzymes [97]. However, to date, mixed functional groupings made up of different subtypes of nAChRs have not been described, nor the possible cooperation or functionality that they would have.

The regulation of the physical interaction between α7- and α3β4*-nAChRs was also investigated. Given the high level of colocalization and interaction of α7- and α3β4*-nAChRs in HCCs, we wanted to understand the mechanisms that regulate this process. It has previously been described in other cellular models that phosphorylation of Tyr residues decreased the membrane expression of α7-nAChRs [98]. To test whether these processes were also mediating the physical interaction with *β4-nAChRs and to ensure that they occurred in native human nAChRs, we repeated the FRET experiment. To do this, we treated the cells this time with drugs specifically targeting enzymes that act on Tyr residues.

By incubating the cells with pervanadate, an inhibitor of the Tyr-phosphatase enzyme, we observed a significant reduction not only in the expression of α7-nAChRs, as already described in other models, but also in α3β4*-nAChRs. This maneuver, which keeps the Tyr residues of the cytoplasmic chain of the nAChRs phosphorylated, also resulted in a significant decrease in the physical interaction.

Treating the cells with genistein, a Tyr-kinase inhibitor, we did not observe an increase in the expression of α7-nAChRs, α3β4*-nAChRs, or the physical interaction between them. However, by incubating this molecule 10 min before treatment with pervanadate, the effect of the latter was significantly reversed, both in the expression of α7- and α3β4*-nAChRs and in the physical interaction between them.

Next, we performed the same type of experiments evaluating the serine/threonine (Ser/Thr)-mediated phosphorylation/dephosphorylation processes using okadaic acid, an inhibitor of Ser/Thr phosphatases. The expression of α7- and α3β4*-nAChRs and the efficiency of their interaction were also reduced. These results suggest that phosphorylation and dephosphorylation processes of Tyr and Ser/Thr residues could also play an important role in the regulation of nAChRs in HCCs, in terms of not only their expression in the membrane but also the regulation of the physical interaction between them.

Regarding sex, we found great differences in the expression of both receptor subtypes. The total fluorescence of BgTx and BuIA in men was higher than in women, indicating that men express a greater number of nAChRs than women in HCCs. Interestingly, both the level of physical interaction and all functional tests of nicotinic currents were not affected by this quantitative difference.

Our data also constitute the first study in which the expression and activity of α7- and α3β4*-nAChRs have been determined differentially in men and women in HCCs. Unfortunately, there is little literature regarding sexual dimorphism in terms of the expression and activity of nAChRs. Historically, the study of females from different animal models is underrepresented in preclinical research, with only 15% of all studies published in the area of neuroscience since 2017 using both sexes for experimentation [99]. Our study shows that HCCs from women express significantly less of both receptor subtypes. This unique finding could be related to the evident differential production of sex hormones, some of which interact non-allosterically with nAChRs [100]. In fact, at the molecular level, neuroactive steroids can act at the intracellular level as transcription factors regulating the gene expression of most ligand-gated ion channels, including nAChRs [101], which would explain the results we observed in our study. On the other hand, the binding of steroids such as progesterone very potently desensitizes nAChRs containing α3 and α7 subunits [102,103]. However, in our study, we show that although the expression of nAChRs is notably reduced in women, they are capable of maintaining the same nicotinic activity that is recorded in HCCs from men.

## 3. Conclusions

Limited research has been conducted on nAChRs in human cells. This article mainly summarizes the most relevant findings achieved in human chromaffin cells obtained from organ donors in our laboratory. We have obtained the result that the main functional subtypes expressed in these cells are the α7- and α3β4*-nAChRs. Maximum-efficiency activation of α7- and α3β4*-nAChRs by the physiological agonist ACh increases the expression of both subtypes in the plasma membrane and prevents their desensitization due to their mutual cooperation. In contrast, maximum-efficiency activation of α7-nAChRs and the partial activation of α3β4*-nAChRs induce the desensitization of the two subtypes. Thus, Chol acts as a potentiation limiter of nAChR activity. Another limiting factor of activity is Ca^2+^ since, in its absence, nAChR activity is not increased after repeated stimulation with ACh. In addition, α7- and α3β4*-nAChRs physically interact. This interaction is independent of the activity of the nAChRs and is regulated by phosphorylation/dephosphorylation processes on Tyr and Ser/Thr residues. Finally, the expression, but not the physical interaction or activity of nAChRs, varies with sex, such that it is significantly lower in women than in men.

## Figures and Tables

**Figure 1 ijms-25-02304-f001:**
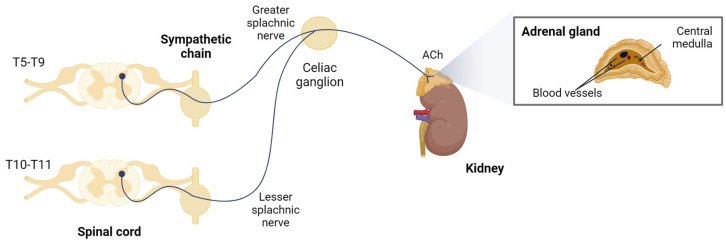
Sympathoadrenal system. Chromaffin cells are innervated by sympathetic preganglionic fibers of the splanchnic nerve. Created with BioRender.com, accessed on 17 January 2024.

**Figure 2 ijms-25-02304-f002:**
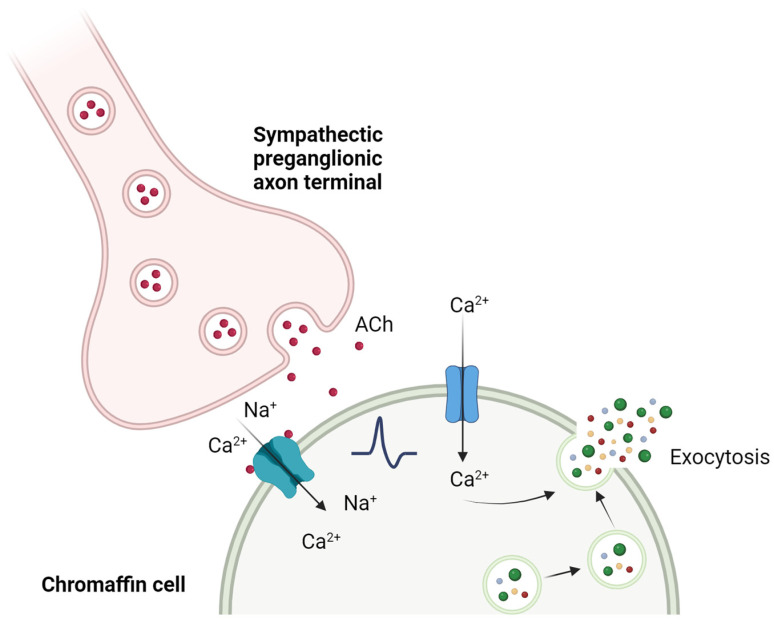
“Excitation-secretion coupling” in the chromaffin cell. The release of ACh by the splanchnic nerve activates a cascade of events, leading to the increase in cytosolic Ca^2+^ and to the fusion of secretory vesicles with the plasma membrane that release their content to the extracellular medium. Created with BioRender.com, accessed on 17 January 2024.

**Figure 3 ijms-25-02304-f003:**
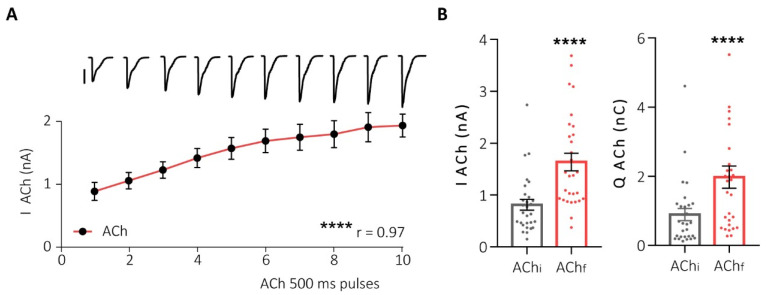
Successive pulses of Ach increase the nAChR-elicited current. (**A**) Representation of the increase in nicotinic current peak in HCCs after 10 Ach pulses of 500 ms/90 s. At the top, representative original recordings of each pulse (500 pA scale). (**B**) Dot plots of the peak (**left panel**) and charge (**right panel**) of the initial (AChi) and final (AChf) ACh pulses in this protocol (potentiation protocol). **** *p* < 0.0001. Taken from Jiménez-Pompa and colleagues [62].

**Figure 4 ijms-25-02304-f004:**
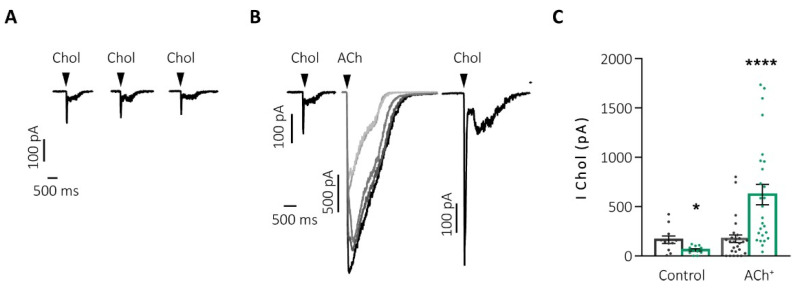
The α7-nAChR requires the activation of the α3β4-nAChRs to avoid their desensitization and to increase their activity. (**A**) Representative recordings of Chol currents after successive stimulation by 500 ms/90 s pulses. (**B**) Chol currents before and after the potentiation protocol by successive (increasing darker shades of grey) pulses of ACh (500 ms/90 s both agonists). (**C**) Graphic representation of the current at the initial (gray) and final (green) peak of Chol after the two previous protocols: control (**A**) and potentiation by ACh (ACh+, (**B**)) * *p* < 0.05; **** *p* < 0.0001. Taken from Jiménez-Pompa [62].

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
