# Peer review of "Nicotinic Receptors in Human Chromaffin Cells: Characterization, Functional and Physical Interactions between Subtypes and Regulation"

_ijms, 2024, doi:10.3390/ijms25042304_

Round 1
Reviewer 1 Report
Comments and Suggestions for Authors
The article "Nicotinic receptors in human chromaffin cells: characterization, interaction between subtypes and regulation" is a review of their previous work on characterizing nicotinic receptors in human chromaffin cells. This article discusses various subtypes of these receptors and their interactions. I have a major concern that this work is mostly a repetition of their recent publication in Journal of Neuroscience [J. Neurosci. 2022, 42(7), 1173-1183]. This article may not be suitable to publish in the International Journal of Molecular Sciences. This manuscript might be more appropriately reclassified and resubmitted as a review article, following significant revisions. I propose the following major modifications for consideration if the manuscript is to be resubmitted as a review article:
1. Expansion of the Review Scope: The article should be restructured to encompass a broader review, integrating results and figures from various research studies, not limited to the authors' 2022 publication. This approach would provide a more comprehensive and balanced overview of the field.
2. Detailed Analysis of Research Variability: The article would benefit from a distinct and thorough discussion of the differences in research methodologies and outcomes between animal and human studies, highlighting the specific challenges encountered in each. The introduction and discussion sections should reflect these distinctions, including efforts by the authors and other researchers to address these challenges.
3. In-Depth Statistical Analysis: The authors have noted gender and age differences in the expression of nicotinic acetylcholine receptor subtypes. A more robust approach would involve conducting a systematic statistical analysis to demonstrate these differences clearly and conclusively. Presenting statistically significant results would substantially strengthen the paper's conclusions and overall scientific contribution.

Quality of English is satisfactory.
Author Response
We have now explained in more detailed the previous work on non-human and human chromaffin cells from other laboratories regarding the characterization of nicotinic receptors and their regulation, and discussed it in comparison with our work on human chromaffin cells.
In addition we have to clarify that the commission received when inviting us to write this manuscript on nicotinic receptors was about our work, which has focused on human chromaffin cells. Therefore, after describing studies reported by other laboratories, the main body of this review focuses on our own results. We have now suppressed experimental details that are not appropriate for a review article and can be found in our original articles.
In relation to age and sex data, we have removed previous reference to age dependence which was a mistake because those data are not published yet. And regarding sex, we can comment data reported in our manuscript by Jiménez-Pompa and colleagues (2022), we do not have any additional analyzes to perform, nor would they be justified being a review article, as mentioned by the evaluator himself in point 1.
Reviewer 2 Report
Comments and Suggestions for Authors
"Nicotinic receptors in human chromaffin cells: characterization, 2 interaction between subtypes and regulation" is a review that summarize all the work of the authors about nicotinic acetylcholine receptors in human chro-9 maffin cells, as the same authors states in the summary. The review is well organized and eloquent in explanation, the initial background is clear and satisfying, allowing a better understand of the experiments reported later on in the text. Also the experiments are well explained, even if sometimes some acronyms are used assuming that the reader is familiar with their meaning. This forced me to check the original paper few times to have the full understanding of the text. I appreciated the critical discussion of the described experiments that illustrates the chronological story behind the authors publications and their rational.
My concern is based on acronyms: DA in line 73 is non explained, the same for ArIB in line 255. It would be better to use the full name the first time you use an acronym to make the text (and experiments) clear to any reader, even the less expert on the field.
Author Response
Acronyms are now explained.
Round 2
Reviewer 1 Report
Comments and Suggestions for Authors
This article, titled 'Nicotinic Receptors in Human Chromaffin Cells: Characterization, Functional and Physical Interaction Between Subtypes, and Regulation,' is a comprehensive and insightful review focusing on nicotinic acetylcholine receptors in human chromaffin cells. The authors have addressed most of the comments from the initial revision.
The manuscript makes a significant contribution to the limited body of literature in this area, particularly by highlighting the characterization of receptor subtypes and the crucial crosstalk between the α7- and α3β4* subtypes to mitigate desensitization. The review is well-structured, detailing the available research on both human and animal studies, along with the authors' own work. I recommend accepting this article after addressing the following minor comments:
1. Typos:
Page #1: Line 24 has a typo.
Page #5: Line 184 and line 190.
Page #6: Line 226.
2. The article often assumes a level of reader familiarity with the terminology. Consider adding more detailed explanations where possible.
3. Figures:
Figure 3: Consider removing the yellow borders for consistency and clarity.
In the legends: Add 'Figures adopted with permission' where applicable to clarify copyright permissions.

Author Response
Thank you very much for your comments. I am attaching the revised version of the manuscript.
